# 'It's not something you can take in your hands'. Swiss experts' perspectives on health data ownership: an interview-based study

Andrea Martani  ,[1] Lester Darryl Geneviève,[1] Bernice Elger,[1,2] Tenzin Wangmo[1]

¹Institute for Biomedical Ethics, University of Basel, Basel, Switzerland
²University Center of Legal Medicine, University of Geneva, Geneva, Switzerland

**Correspondence to**
Andrea Martani;
andrea.martani@unibas.ch

## ABSTRACT

**Objectives** The evolution of healthcare and biomedical research into data-rich fields has raised several questions concerning data ownership. In this paper, we aimed to analyse the perspectives of Swiss experts on the topic of health data ownership and control.

**Design** In our qualitative study, we selected participants through purposive and snowball sampling. Interviews were recorded, transcribed verbatim and then analysed thematically.

**Setting** Semi-structured interviews were conducted in person, via phone or online.

**Participants** We interviewed 48 experts (researchers, policy makers and other stakeholders) of the Swiss health-data framework.

**Results** We identified different themes linked to data ownership. These include: (1) the data owner: data-subjects versus data-processors; (2) uncertainty about data ownership; (3) labour as a justification for data ownership and (4) the market value of data. Our results suggest that experts from Switzerland are still divided about who should be the data owner and also about what ownership would exactly mean. There is ambivalence between the willingness to acknowledge patients as the data owners and the fact that the effort made by data-processors (eg, researchers) to collect and manage the data entitles them to assert ownership claims towards the data themselves. Altogether, a tendency to speak about data in market terms also emerged.

**Conclusions** The development of a satisfactory account of data ownership as a concept to organise the relationship between data-subjects, data-processors and data themselves is an important endeavour for Switzerland and other countries who are developing data governance in the healthcare and research domains. Setting clearer rules on who owns data and on what ownership exactly entails would be important. If this proves unfeasible, the idea that health data cannot truly *belong to* anyone could be promoted. However, this will not be easy, as data are seen as an asset to control and profit from.

## Strengths and limitations of this study

► This study deals with health data ownership, an important topic that lies at the crossroads of health policy, medical informatics, bioethics and law.

► It offers a qualitative approach to analyse consistencies and inconsistencies in how experts perceive ownership and control of health data.

► The findings of this qualitative study are not generalisable and—due to the semi-structured nature of the interviews and the limited time for probing questions—some of the topics would require further investigation.

## INTRODUCTION

As healthcare is set to become an increasingly data-rich environment, the question of how to govern the medical information collected and processed represents a great challenge.[1] Appropriate governance is especially important at a time when initiatives to promote the sharing of data between countries and institutions are growing. For example, in the context of the Coordinated Research Infrastructures Building Enduring Life-science Services European project, there have been efforts to establish solid principles to guide the (re)use of individual patient data from clinical trials.[2] Similarly, the discussion on how to properly govern the exchange of data between different actors has been prominent also in the USA.[3] Even single research institutions—like the UK Biobank—have dedicated particular attention to the ethics and governance of the data they manage.[4] Indeed, failing to establish appropriate and socially accepted governance for medical information can lead to significant problems.[5] For example, the Care.data project of the National Health Service (NHS), which was aimed 'to extract data from NHS primary care medical records in England unless patients have purposefully opted out, in part to facilitate research',[6] proved deeply controversial also due to some fallacies in its governance.

One very important issue in the governance of health data is determining whether

ownership in data exists, whom it refers to and what exactly it is.[7] Due to the lack of a commonly accepted definition, the debate on data ownership focuses on two sides of the issue: rights to *control* data and to *benefit* from them.[8] Such debate is present in the biomedical research community,[9] in the ethical domain[10] and especially in the legal field (see eg, this review of the literature).[11] In this respect, it is often questioned whether and how data can truly 'be owned' in a legal sense,[12] and whether it is desirable to extend norms concerning other kinds of property to data.[13] Although originally focused predominantly in the USA, the debate on the 'propertisation' (ie, the application of property-like rules) of personal data has then progressively expanded in Europe as well.[14] This has also been fuelled by the reform of data-processing rules in the European Union brought about by the *General Data Protection Regulation* (GDPR). Indeed, the GDPR represented an effort to better define the role and powers of individuals where data come from (data-subjects, art. 4 (1) GDPR) and of those who manage data collections (data controllers and data processors, art. 4 (7), (8) GDPR). Some have described the GDPR as a decisive step towards the implementation of a property regime for data[15] and others have been more sceptical about this.[16]

The idea of introducing property or ownership rights in data concerning health has received particular attention. Critics note that without clear and transparent rules about ownership of patients' data, an uneven level-playing field has emerged, where the use of data in medical research is overregulated, and use of health data in the private domain is underprotected.[9] In this respect, ownership by patients has been pointed as a potential way forward to benefit both individuals and healthcare in general.[17] Moreover, clear ownership rules concerning medical data have been proposed as one important step to capture the benefits that data can produce in healthcare.[13] Concerns have also been expressed that granting real 'property-like' rights in their data to patients might hinder important research, due to the excessive control these rights would give to individuals.[18]

The attention to the topic of health data ownership on a theoretical level is also mirrored by the increased scrutiny that this issue has received in empirical literature. In this respect, research has considered mostly the questions of who should be the owner of health data[19–25] or underlined that there is lack of (legal) clarity about rules and rights concerning data control.[26–31] However, existing empirical research discuss this issue from a broad perspective in the context of data management and data reuse, without focusing extensively on different aspects of data ownership and its meaning.

In this paper, we explore specifically the topic of data ownership in the healthcare domain based on interviews with Swiss experts involved in the processing and sharing of medical information. Switzerland makes no exception to the international tendency of harnessing the benefits that digitalisation and the use of data in healthcare and for biomedical research can bring about. For example,

in 2020 a new decentralised system of interoperable electronic health records started being operative[32] to permit citizens to have more control on their data and to share them with all the medical providers from whom they receive care. Moreover, some data-sharing platforms that 'enable citizens to be in control of the storage, management and access of their personal health and health-related data' have been introduced in the country.[33] The topic of data ownership has also started being debated in the legal field.[34 35] In consequence, understanding ownership with respect to medical data is particularly relevant. The current paper is part of a broader project on the topic of health data harmonisation,[36] where we considered experts' views and concerns related to the collection and sharing of health data in Switzerland, as well as its legal, organisational and ethical implications. In this paper, we only report findings on the topic of data ownership.

## METHODS
### Ethical considerations
Structure and objectives of the study were illustrated to the prospective participants when contacted and then described again before each interview. Participants orally agreed to take part in the study, that their interviews be recorded, transcribed and used for the project after personally identifying information was eliminated. On request, transcripts were returned to participants to correct them and eventually ask for the elimination of some segments, whereas checking of the findings was not planned for this part of the project. We followed the COnsolidated criteria for REporting Qualitative research reporting guidelines.[37]

### Study design
We conducted interviews with 48 experts involved in the processing, governance or collection of health-related data in Switzerland. For the study, a total of 58 experts were contacted: 10 either declined the invitation or did not reply. Experts were interviewed either in person (n=36) or via phone/skype (n=12) based on their preference. Most of the interviews (n=39) were one-to-one. A few experts requested to be interviewed with members of their team for practical reasons. Thus, nine participants were interviewed either one-to-two or one-to-three.

### Sampling
We relied on purposive and snowball sampling. An initial list of potential interviewees from the three experts' groups was elaborated from the studies considered in a systematic review conducted previously by our team.[38] The objective of our sampling strategy was to include both experts on the topic of health data from a more practical side (eg, researchers, hospital directors) and those with a more institutional perspective (eg, policymakers, directors of health registries). Often, experts had (or had previously had in their career) more than one role. Interviewed participants were asked to provide recommendations so

that our team could contact other experts from the Swiss context. Our sample did not include patients or members of the public, since we wanted to focus on stakeholders with first-hand experience in the management and use of medical databases. To indirectly tap into the perspective of patients, we had also contacted an individual with experience as patient think-tank director, who however declined our invitation for lack of specific expertise on the topics.

## Data collection

A semi-structured interview guide was developed by the study team and pilot-tested for content and understandability. The interviews were conducted independently by either LDG or AM, two male PhD candidates with training on data collection and qualitative research methods. During one interview by AM, another PhD student came along as an observer to gather experience on the conduction of interviews, on agreement of the interviewee. As a default, interviews were conducted in English, but the other three major official languages of Switzerland were also offered as an alternative, with some participants opting for Italian (n=4), German (n=3) or French (n=1). The interviews took place between May 2018 and September 2019. In one case, a participant asked to integrate the interview with a second conversation. The recordings lasted between 38 and 131 min, with a median duration of 60 min. Recordings were transcribed verbatim, integrated with field notes and potentially identifying information— for example, age, exact place of work and position within the organisation/institution— was eliminated to ensure confidentiality. Overall, 10 participants were directors or managers of health information databases/registries, 28 were researchers working with health data in project of national relevance and 10 were experts from the public administration involved with health data management. Data collection was stopped when no new issues were emerging in the interviews and it was thus deemed that data saturation had been reached.

## Data analysis

To analyse the data, we relied primarily on thematic analysis[39] and were guided by the framework provided by Hansen.[40] In our analysis, we also followed the recommendations by Silverman,[41] in particular by considering both a positivistic (ie, the idea that the content of responses conveys primarily an external 'truth') and an interactionist perspective (ie, the idea that answers also mirror the situational contingencies where they are given).

Our analysis started after the first interviews were conducted. Seven interviews were read and discussed together by AM, TW, and LDG during a series of meetings and a preliminary coding tree with themes and subthemes was developed. We used MAXQDA.18 as an analytical software to help with our analysis.[42] Afterwards, the remaining interviews were analysed individually by AM and LDG relying on the previously developed coding tree. Thereafter, a series of meetings were organised to discuss additional 15 content-rich interviews to refine the coding tree, its themes and subthemes and to develop key findings from the data.

As this study is part of a broader project aimed at tackling several issues related to the health data framework and infrastructure in Switzerland, several broad themes and codes were identified during the data analysis. These included, for example, recommendations on how to improve health data infrastructure in Switzerland and how to promote fair data sharing practices. For this paper, we relied only on the broad theme of data ownership, which emerged when asking questions about the barriers to the acquisition and sharing of health data, about the presence of regulatory barriers or about the reasons why data-processors are hesitant to share medical data they collect. All segments related to the theme of data ownership were extracted and a topic-specific coding tree was developed, leading to the classifications presented in the results. All authors reviewed, edited and approved the specific coding tree for data ownership and participated to the analysis of the results.

## Patient and public involvement

Patient and public involvement was not part of this study. See section 'Sampling' for further details.

## RESULTS

Four main themes concerning data ownership were identified, as presented in figure 1.

### The data owner: data-subjects versus data-processors

Participants often reflected on the subject of whom the ownership of data can (or should) be ascribed to. In this respect, two main potential data owners were mentioned, namely the data-processors (ie, the institutions or individuals, often researchers, who collected and/or used the medical information) or the data-subjects (ie, the patients or, more generally, the persons to whom the health data refer). In our results, the term 'data-processors' is used *in a general sense* to indicate all those institutions/individuals who practically collect and/or manage the data for a certain purpose, as opposed to data-subjects (ie, the individuals—often patients—where the data comes from). According to the GDPR, there can be a distinction between those institutions or individuals that actually carry out the processing, and those which determine 'the purposes and means of the processing' (art.4 (7) GDPR), which are named 'data-controllers'. We did not consider this distinction for our analysis, since it was not present in Swiss law at the moment when interviews were carried out. The distinction between data-controller and data-processor will be introduced in the near future by the reform of the Swiss Federal Act on Data Protection, which will come into force in 2022 and is referenced in the 'Discussion' section.

Some experts (generally researchers) leaned towards associating ownership with the data-processors. In E1

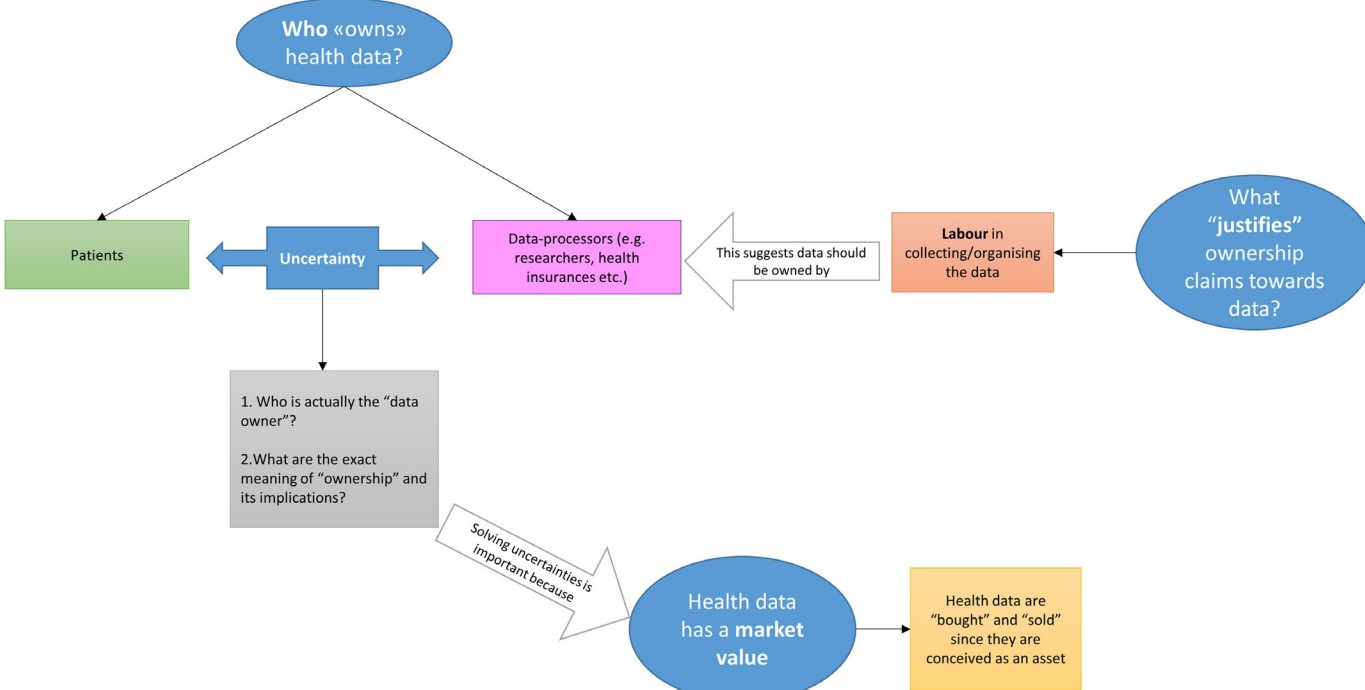

**Figure 1** Representation of the themes related to health data ownership.

(table 1), for example, an expert talking about a project involving the use of medical information provided by an insurance company explained that the data belonged to the insurance company. Interestingly, experts suggesting that ownership pertains to data-processors were subtler in expressing this connection, as compared with those who argued that patients are the owners. Indeed, the former tended to speak of data-processors and data as being 'theirs', rather than using specifically terms like 'ownership', 'property' or 'belonging to'. The director of a health database (see E2 in table 1) explained that the medical centres contributing data to the database considered such data as 'theirs', but did not mention the concept of 'belonging'.

Other participants (including also researchers, but mainly experts involved in health registers or other health data infrastructures) indicated that health data fundamentally belong to the data-subjects, that is, the patients or more in general the citizens from whom information is collected. As compared with those participants who sided for data ownership by the data-processors, those who argued that data are owned by patients often used firmer language (E3 in table 1). The latter, despite their firm claims that data belong to patients, tended not to justify such claims—as if the fact that data belong to patients is self-evident and needs no further justifications. This is very different as compared with the ascription of ownership to data-processors, for which an explanation was provided (see 'Labour as a justification for data ownership' section). Only some experts seemed to suggest that ownership by the patient has to do with 'empowering' them (E4 in table 1).

Even those who ascribed ownership to patients seemed to be aware of the fact that—traditionally—the processors of the data (eg, doctors, hospitals or health insurance companies) feel entitled to data ownership by default. For example, one expert expressed this idea in E5 (table 1) and argued that this tendency has to be changed. One participant who was in favour of ownership by patients indicated that one of the reasons why data-processors might prefer to consider health information as their own property is that they see data as an asset (E6 in table 1). Another participant argued that an additional reason why data ownership by patients is opposed by data-processors (in that case doctors) is the paternalistic conviction that patients would not be able to deal with information about their own health (E7 in table 1).

Although in the literature the idea of common data ownership or public ownership of health data has been advanced,[43 44] only two participants showed some sympathy for the idea that data should be open and treated as some kind of *common property*. One participant (23REngP) spoke about it with reference to data collected by researchers (but see also E1) and one of them (18PGerP) with reference to data collected by governmental agencies.

## Uncertainties about data ownership

Whereas many experts sided for data ownership either by the patients or by data-processors, another theme often discussed was the uncertainty related both to 'who is the data owner' and 'what is the meaning of data ownership'. In an exchange during the only one-to-three interview, it emerged how the uncertainty as to the question 'who is the data owner' is related to a tension that exists between

| Table 1 | Extracts on the topic of assigning data ownership |
|---|---|
| **Extract number (E)** | **Quotes from participants** |
| E1 | "And I know that the (name of funding agency) now requires that all data should be on some platform.[…]. But we cannot give the data, because they belong to (name of a health insurance)." 23REngP* |
| E2 | "Because it is possible that every now and then one center was not keen on transmitting **its** data because it has/because it preferred keep them for itself and publish them in another way. We experienced this. There were a couple of centers which preferred to keep **their** data[…], which told themselves: 'But we have already enough data so that we can publish them'[…]But then I have to say that even this resistance is—typically—an initial resistance. Because some centers say 'Wait a minute: Why do we need to make **our** data available to you, who then publish them?' (emphasis added)." 8HItaT |
| E3 | "You start actually with patients' data. So the data—first of all—belongs to the patient. […] Meaning, also for a doctor, or a researcher, the data does not belong to these people, the data belongs to the patients." 43HEngP2 |
| E4 | "One solution is the empowerment of the patient or the person[…]And in these in [year] I have developed [name and description of a data collection system]. And this system works in the sense that it is the patient that actually is the ultimate owner of the data[…]. So basically you must have a system where the data are being owned by the patient rather than by the institutions." 2REngT |
| E5 | "[B]ecause the institutions or the head of the institutions they feel the data belong to them and you can only give it to somebody with trust. And first of all you have to explain to them that the data don't belong to them." 21HEngP |
| E6 | "So, they [data-processors] are not interested in sharing. They are not interested in sharing now, because they see these as a sort of… all the medical data as sort of an asset." 27HEngP |
| E7 | "They [the doctors collecting data] say: 'It's my data, you know, I wrote this data. Sometimes there's things in these reports that the patients should not know or should not read.' So this understanding that you take your patients seriously as persons who also can deal with difficult topics if you do it the right way. This is a cultural change and this is not easy to establish." 11PEngP |

Continued

| Table 1 | Continued |
|---|---|
| **Extract number (E)** | **Quotes from participants** |

*The abbreviations after the quotes give details about the interviewee. The first number is the number of the interview, assigned chronologically according to date of conduction. The first letter refers to their positions: **R** = Researcher, **P** = Policymaker or public official; **H** = High position in a health register, IT infrastructure or a relevant initiative concerning health data. Then, the original language of the interview is indicated: **Eng**= English, **Ger** = German, **Ita** = Italian, **Fre** = French. Thereafter, the medium with which the Interview was done is indicated: P = in person, T = on the phone, S = on Skype. If a number is present at the end, it means that two or more experts participated to the interview and the number indicates the expert from which the quotation comes – unless otherwise indicated. For example 23REngP means: 23rd interview, with a Researcher, conducted in English and in Person.

the fact that health data are related to patients, but the latter are often not practically in control of what happens with them (E8 table 2). Uncertainty concerned also the question of what exactly the meaning and the features of ownership are. One participant, for example, explained how difficult it is to apply categories like 'property' and 'ownership' to an incorporeal item like data, since those categories are traditionally conceived for tangible objects (E9 in table 2). On a similar line, the complexity of understanding what the concepts of 'belonging' and 'ownership' actually mean when applied to data was also mentioned as a source of uncertainty (E10 in table 2).

## Labour as a justification for data ownership

As different individuals may potentially make ownership claims towards the same health data, participants often reflected on what entitles to owning the data. In this respect, there was a tendency among participants to express a strict connection between labour and property, with the idea that putting effort and energy to collect and manage health data legitimises ownership claims towards them. This connection between labour and data ownership was sometimes expressed in a very direct and strong fashion. Interestingly, even some participants who argued in favour of data ownership by the patients acknowledged (although with some criticism) that there is a widespread sense—especially in the field of research—that having put some effort in collecting and organising the data renders the data 'yours' (E11 and E12 in table 3). The contrast between the conviction of some participants that data have to belong to patients—on the one hand—and the fact that data-processors feel some kind of ownership due to the effort they put in organising the data was well highlighted also in another interview (E13 in table 3).

In other cases, participants provided a more differentiated description: they did not claim explicitly that labour with respect to data entitles to be the data owner, but still creates some 'privileges' with respect to those data. In one passage, an expert referred to the fact that the work for preparing the data and passing them on should be

**Table 2** Extracts on the topic of uncertainty

| Extract number (E) | Quotes from participants |
|---|---|
| E8 | Participant 3: "However, this thing concerning health insurance funds actually… 'whom does the data belong' is really a question that I ask myself. Whom do they belong to, since the patient has given them and they are anonymous…to whom do they belong?" Participant 1: "They belong to the patient probably. However, the problem is that the availability of data does not depend on the patient. Because the patient does not have her own data…That's it." Participant 2: "Because then it is the 'data management'—do you understand?—who takes care about them [the data]…I authorize you to process my data, but if I am then not the one who manages or processes it…" 14RItaP |
| E9 | "[A]t different conferences, meetings this [data ownership] is always a discussion point, but it doesn't come to a solution. It ends open mostly. And I think it's, yeah… it's difficult because you have to find out what is data. It's not something like a cup or something you can take in your hands, it's not material in that perspective. So can you own it or can you get only the right to process it? It's very, very tricky." 21HEngP |
| E10 | Participant: "I think this 'belonging' of data is somehow a difficult story.[…]Maybe it is legally at first sight straightforward, but on second thoughts it is—I think—more complicated." Interviewer: "Well, theoretically yes [it is easy] but practically not so much…" Participant: "Yes, right? And what does it mean actually 'they belong to me'?" 36PGerT |

**Table 3** Extracts on the topic of labour as a justification of data ownership

| Extract number (E) | Quotes from participants |
|---|---|
| E11 | "If you [as a researcher setting up everything for data collection] have the relevant data, the IT-platform, if you have all the regulations in place… you know, in [name of one project] I had to invent all of that. We do everything from 'A' to 'Z'. So, then of course you have some ownership." 38REngP |
| E12 | "[T]here is too much about…it's too much often protection of your own work, of your own project, also because you have actually gained the funding—so why should I give my data which I collected by my funding and so on.[…]. It is a very, it is a narrow way of thinking." 43HEngP2→ supporter of patient data ownership, but acknowledging how data-processors justify making ownership claims. |
| E13 | "[B]efore—[name of P's project]some institutions and researchers said: 'These data belong to me.' They have built a biobank from patients and so. Maybe they have 200 samples or something like that in research. And then they say: 'This biobank belongs to me.' Then you have to tell them: 'No, no, no, no. Just stop. Nothing belongs to you. All the data belong to the patients and not to the institutions.' But for sure—I mean—the institution invests money, and so the institutions think: 'Oh we cannot just give out the data, otherwise we have invested some money and we want to have the investment back at least'." 39HEngT |
| E14 | "I'm happy that you used our data, I don't need to be an author because I did not contribute as an author, but I would like to be listed as a collaborator and this is what they do. So I think this is a very good way to acknowledge and you know nowadays I mean authorships are nice to have, but it's much more important to show to others that you are participating and you have a strong network." 26REngS |
| E15 | "There would have to be agreements you know… if I let you see these data, you know, are you going to give credit to the people who worked on them you know…who actually did the work of getting them[Interviewer: collecting?]…Yeah, yeah… That is something that has to be worked out… that's a lot of work you know." 1REngP |

somehow acknowledged, also in other ways than simply receiving authorship. Also, the idea of having invested time and energy in collecting or preparing health data for analysis was sometime mentioned as generating not exactly ownership, but rather some kind of 'pre-emptive' rights—that is, the right to have a say before the health data are forwarded or used for something else (E14 and E15 in table 3).

### Data have a market value

In the literature, data ownership is often discussed in relation to the development of an efficient market for data.[45] In particular, it is claimed that clarifying who the data owner is and what exact powers ownership entails would facilitate the trading of data. Despite the diverging opinions concerning who is the data owner and the uncertainty as to what powers ownership entails, in our interviews data were often discussed in market terms. Some participants, for example, referred to how data are now conceived as a commodity and monetised, or alluded to a health data

market when discussing how data are exchanged (E16 and E17 in table 4). One participant was quite critical of the position that giving ownership and control of their data to citizens will empower them to become active participants in the data market and trade their medical information for money (E18 in table 4). The tendency to think about

**Table 4** Extracts on the topics of data having a market value

| Extract number (E) | Quotes from participants |
|---|---|
| E16 | "And second, probably there is now the consciousness that data are a resource. So you can make money with data, and so it's better to…well it's better to make some market with your data than just sharing them. Because we are all in a very… big pressure for resources and so if you can have…if we can make some money with the data, we can get some more resources to work." 28REngP |
| E17 | "I mean the problems you realised at first hand by seeing how everyone is very protective of what they think is a great asset. Because, you know, I mean in the US [name of a pharma company] just paid 1.5 billion dollars for (name of a company), which is a company that bought—bought! [emphasising the word]—a million hospital record and annotated them a bit." 27HEngP |
| E18 | "I have heard some people say: 'Oh wow, as soon as I have data, I will share this data—for instance—with industry, because industry is looking for data and they will pay for this data and then we can make money and we can share this money with people who share their data with us […]. I think it's ridiculous. It's a very poor understanding about industry." 30REngS |
| E19 | Interviewer: "We usually hear that researchers are not very keen in sharing the data that they've collected." Participant: "Yeah. […] I mean, researchers, they want to share data usually as long as they receive data from others." 33HEngP |
| E20 | "I think quite a lot of people are—how would I say that…some kind of 'want to do good'…if it comes to research, you know, if your research can explain why he is doing something and, let's say, what the purpose is and how it can help—so to say—the system…most of the people will, let's say, donate the data." 24HEngP |

health data as a tradeable commodity was also present in some participants who spoke about data-exchange as a sort of 'barter' between researchers (E19 in table 4).

There was only one interviewee who explicitly contrasted a market language with respect to data-exchange and referred to the fact that people might be motivated by altruism when they provide their health data, thus 'donating', rather than 'selling' them (E20 in table 4).

## DISCUSSION

In this paper, we reported the results from our interviews with Swiss experts concerning the topic of health data ownership. In this respect, our study findings underline very diverging convictions concerning who is the data owner, possibly also because there is some uncertainty about what ownership entails. Other qualitative research had suggested that ownership of data resides either with the patients[22–24] or with the data-processors, such as local or public health authorities,[20 21 30] but did not investigate how to reconcile these two opposite claims. The presence of these two competing conceptions might be explained with the indeterminacy of the concepts of 'ownership' or 'belonging'. In this sense, it could be that experts siding for patients' ownership referred to the fact that data are *about* patients and *provided by* patients, and thus *theirs*. On the other hand, our study participants claiming that data-processors own the data might have thought of ownership in terms of 'having something at your disposal', which applies—with some limitations—to the relation between data processors and the data they have. Indeed, our interviews also highlighted confusion about the meaning of ownership (see discussion below). Alternatively, the divergence of opinions could be explained by the fact that participants were making *normative* (ie, indicating how things *should be*), rather than *descriptive* (ie, referring to how things *are*) statements. The idea of health data being conceived as a common good or a public property—which has been discussed by some authors[43 44]—was seldom mentioned. This might be due to the specific features of the Swiss context, where there is no national healthcare system,[46] and thus the de facto control over many resources in healthcare (including data) is divided between a multitude of stakeholders, rather than hold by one public entity.

Our results confirm that there is lack of agreement about who the final data owner should be, and uncertainty about what *ownership* exactly entails. This finding is particularly important, since it suggests that experts—although often referring to the idea of *ownership* of health data—are not always certain what this concept means. In general terms, ownership can be defined as 'a set of relationships, granting to a person or to a group control over a specific resources vis-à-vis other people',[47] but this term assumes different connotations depending on the context in which it is used (eg, legal vs economic). The meaning of *ownership* as applied to *data* is even more complicated, since claims of *data ownership* can be legal, and also have a political and philosophical scope.[48] One straightforward reason why data ownership is a complex concept is that data possess a rather intangible nature, as compared with other objects of ownership rights, and applying ownership to immaterial objects has been controversial.[49] Legal experts are indeed still very sceptical about using the (legal) concept of ownership to describe the relationship between data and the subjects that interact with them.[50] In the Swiss parliament, there was an effort to modify the Constitution to declare personal data as *property* of the individual,[51] but this modification was rejected. On the contrary, the recently reformed *Federal Act on Data Protection*[52] recognises health or genetic information as a

sensitive kind of data, but it does not grant any ownership rights of a legal nature towards them. Our interviews confirm that the idea of data ownership has a certain appeal, but that there is still great uncertainty as to the exact meaning of this concept. It is possible that, if prompted with more concrete cases, our participants would have expressed less uncertainty about what they conceive as *ownership*. Nevertheless, since this concept is often used to described the control of people (be them the patients or the data-processors) over health data, its meaning should be clarified a priori. Better defining the meaning and extent of ownership entitlements has also been described as an important condition for the data market to thrive.[45]

Moreover, our study indicates how there is a widespread conviction that labour provides a justification for owning the data (or dataset) for which the work was done. This would lead to think that ownership of data should reside with the data-processors and not with the patients—since the former do most of the work to collect and use the data. The idea that labour may be used as a justification for ownership claims was also shared by some experts who—in other part of their interviews—had suggested that patients should be owners (eg, interview 43HEngP). On the contrary, no explicit and extensive justifications as to why patients should own their data were provided by those who argued in favour of this. When considering the views of supporters of patient data ownership in the USA, Evans[53] explains that normally these rely on arguments of control and of full access to one's own information. It is possible that similar motivations drove the statements of the experts in our study. In Swiss legal literature, both the possibility of assigning ownership to data-subjects—because the data originate from them—and to data-processors—because they invest in the collection, storage and use of data—have been discussed.[34] The claim that patients should own their data might be popular also because Switzerland has a long tradition of strong individual rights and citizens' control, as its political system is characterised by direct democracy (eg, with referenda and popular initiatives). Furthermore, Switzerland is host to two of the first international examples of *data-cooperatives*, that is, data-sharing platforms strongly leaning towards the idea of citizens' control of their data.[54] Data-cooperatives are databases 'concerned with the collection, storage, maintenance and analysis of health data' where every citizens can participate by 'pay[ing] a one-time unit price (membership fee), which entitles the person to be a member and owner [of the cooperative] at the same time'.[55] The existence of such initiatives in Switzerland corroborates the idea that patient ownership and control over health data is acknowledged. Nevertheless, it is surprising that no justification of the claim that patients are data owners was expressly present in our interviews. One obvious reason that would support the claim that patients are to some extent 'data owners' is the fact that they must often provide consent before their data are used—thus exercising some sort of control

on the data. Indeed, informed consent of the data-subject (ie, the patient) is considered a cornerstone of legitimate data processing from an ethical and legal point of view.[56] Both Swiss law and the GDPR indicate that obtaining informed consent by data-subjects is often necessary to permit the processing of medical data, which are considered as 'particularly sensitive' (art. 3 of the Swiss Federal Act on Data Protection and art.9 GDPR). However, it is also true that there are many cases where data processing might proceed without patients' explicit consent and it is justified by a different lawful basis for data processing.[57] For example, if medical data are sufficiently anonymised (there are doubts, however, whether data nowadays can ever be considered as truly anonymised, especially with respect to genetic data[58]), patients' consent becomes generally irrelevant when using such data (eg, for public health purposes).[59] Or else, even if the data are *not* anonymised, regulation often caters for 'research exemptions',[60] that is, specific rules that allow the secondary processing of medical data for research purposes (eg, the use of data initially collected during clinical routine to then conduct healthcare service research) without the need of the explicit consent by patients (see Martani *et al.*[61] for Switzerland). Thus, although it is true that expressing consent for data processing could be a powerful justification to claim that patients 'own' their data, it is also true that the processing of medical data does not always necessitate consent. Despite the existence of established paths to use patients' data without their permission, supporters of patient data ownership in our interviews presented their stance (ie, data belong to patients) in an assertive fashion—hence also the use of a 'strong' and 'resolved' language and the denouncing of the opposite position (ie, data belong to data-processors) as mistaken.

The fact that putting labour with respect to data was described as legitimising ownership claims towards such data mirrors one established philosophical and jurisprudential account of ownership, according to which by commingling labour with an object, one establishes property of it.[62] Such finding is important because it reveals that those working on acquiring or processing health data(sets) develop a sense of entitlement towards owning the data. This sense of entitlement might curb the willingness of healthcare researchers and hospitals to share data and might generate controversies between individual data-processors (eg, researchers working at a hospital), who do the actual work for organising the data, and the institution by which they are employed, which provides the financing and the means. Moreover, if it is perceived that labour gives some degree of ownership over the data (or at least have some pre-emptive rights towards them), the rhetoric of 'patient data ownership'[17] might clash with a reality where those who put an effort to collect patients' data feel they deserve to control that medical information. The need to resolve the tension between the willingness to facilitate data sharing and the feeling of data-processors of having special claims towards the data they manage was also explored by a previous study

conducted with stakeholders from Africa.[63] Our study suggests that a similar attitude is present also in a wealthy country like Switzerland, where resources to collect and manage data are comparatively larger. If health data-sharing and reuse should be facilitated, there might thus be a need to reduce this feeling of entitlement that data-processors perceive, by diminishing the competition for the resources needed to create databases. This could be achieved in different ways. For example, shared priorities concerning the health data that need to be collected and managed could be elaborated at an interinstitutional level, to then distribute the effort necessary for collection and managing of such data between a multitude of actors and thus sever the sense of entitlement of single institutions. Or else, clear guidelines could be developed that explain how to access health data collected by the effort of single data-processors (eg, hospitals) in exchange for a reasonable fee (especially since data are already seen as having a market value), but also without leaving too much discretion as to which access-requests to accept and which not. Needless to say, it is difficult to strike a balance between providing an adequate reward to data collectors and keeping data accessible, but this seems to be the right way to go.

Lastly, our results indicate that using a market language with respect to health data is already quite widespread. This confirms what theoretical literature has been suggesting for some time, namely that health data are often conceived as a commodity.[64] Commodification can be considered ethically problematic, since health data—like characteristics derived from biological material and organs—are something intimate and connected to human dignity.[65] Some authors claim that the dangerous tendency of *commodifying* data can be accelerated, if *property* rights towards them are reinforced (the so-called 'propertisation of data').[66] However, whether having well-defined ownership rules concerning data really increases an alarming tendency to commodify them remains to be ascertained. Our interviewers often spoke about data in market terms and as a commodity, even if in Switzerland (like in any other European country) data are not formally governed by property law and ownership of data remains ill-defined (see section 'Uncertainties about data ownership'). Purtova argued that it is actually the lack of explicit ownership rules that favours the development of a more unbalanced data-market, since 'maintaining the status quo where no ownership in personal data is formally assigned equals assigning ownership to the Information Industry and leaving an individual defenceless'.[67] Although data can be re-used over and over again, and shared with a massive number of parties at the same time, the fact remains that 'the tricky part is in getting to the source of (data), that is, people, in the first place'.[67] A potential way to escape the market logic in data sharing and exchange could be to favour the idea of health data as a common good, which should be simply guarded by the data-processors who actually manage the data themselves.[2] A political initiative in this sense was promoted in the Swiss Parliament during the COVID-19 pandemic, with an appeal by some politicians to create the possibility for citizens to voluntarily donate their health data in the public interest, for both research purposes and public health emergency response.[68]

## Limitations

This study has some limitations. First, the interviewees were experts with often limited amount of time, thus we had to adapt the study design to their needs. This led, for example, to having some one-to-two interviews and a one-to-three interview, to having to conduct some interviews in person and some via phone or Skype and to having limited time for probing questions. Second, the two data collectors have different backgrounds (AM in law and LDG in medicine), thus potentially introducing some variations in the different interviews. Third, as a qualitative interview study, we cannot exclude that responses may have tended towards socially desirable answers. Finally, ours results cannot be deemed a generalisation to what other health data experts in Switzerland and elsewhere believe about data ownership.

## Conclusions

In an increasing digitalised medical sector, data ownership remains a highly controversial topic. This study confirms that uncertainty still surrounds both the question of who can be considered data owner and what 'ownership' exactly entails. The fact that labour was mentioned as a reason to assign ownership to data-processors (eg, researchers) suggests that the latter feel a sense of entitlement towards the data they manage. This clashes with the opposing claim that data should be owned by patients. The contrast between the rights of patients towards their data and those of data-processors who manage them is typical of several countries (see Rodwin[69] for the USA). The findings of this study may thus be of use for Switzerland, and for other countries which are trying to implement the most appropriate governance to settle this potential contrast. Indeed, there is an international endeavour to implement a form of data governance that continues protecting patients' rights and privacy towards their data, and favours the reuse of data in biomedical research and public health. More precise definitions of data ownership rights of the different parties involved in data collection and their management would be a step forward in this direction. This would entail providing clarifications of who controls data in the different data processing activities and should thus be consulted, for example, when data are shared with third parties. If the language of 'ownership' proves impracticable for the research sector or the use of medical data more in general, then there should be an effort to reject the idea that medical data *belong to* anyone. An alternative could be to favour the idea that these are a sort of public good, which can be *donated* by patients and guarded by data-processors, who would 'have a responsibility to ensure the data are discoverable by others and accompanied by sufficient metadata

for them to be found easily, understood in context and used appropriately'.[2] Yet, this would also require to eradicate another tendency that our study confirmed, namely the fact that medical data are seen as an asset with market value, which—as any other resource—many parties have an interest to control and profit from.

**Acknowledgements** AM would like to thank Patrik Hummel for the profitable exchange of ideas on the topic of data ownership and for the feedback on one of the versions of the manuscript. Moreover, the authors would like to thank the reviewers for their helpful comments.

**Contributors** BE and TW conceived the study and prepared the interview guide with LDG and AM. AM and LDG conducted the interviews. All authors participated in the analysis of the data. AM and TW prepared the first draft of the manuscript. LDG and BE integrated the initial draft with comments and additions to the manuscript. AM finalised the last version of the manuscript, which was then corrected, read and approved by all authors.

**Funding** AM, LDG, BE and TW acknowledge the financial support of the Swiss National Science Foundation (SNF NRP-74 Smarter Health Care, grant number 407440_167356).

**Disclaimer** The funder had no role in study design, data collection and analysis, decision to publish or preparation of the manuscript.

**Competing interests** None declared.

**Patient and public involvement** Patients and/or the public were not involved in the design, or conduct, or reporting, or dissemination plans of this research.

**Patient consent for publication** Not required.

**Ethics approval** According to the Swiss Human Research Act, this study does not require ethical approval. This was confirmed by the cantonal ethics commission (Ethikkommission Nordwest- und Zentralschweiz (EKNZ)), to which the authors submitted the outline of this study and which stated that it fulfilled general ethical and scientific standards, posed no health hazards and did not require formal approval by them (EKNZ req-2017-00810).

**Provenance and peer review** Not commissioned; externally peer reviewed.

**Data availability statement** Data are available on reasonable request. Interviews have been conducted under the assurance of confidentiality concerning the identity of the interviewed experts, hence potentially identifying information has been masked during transcription. Due to reasons of confidentiality, full transcripts cannot be shared—as the potential of re-identification would be relevant. Data and segments used for the following manuscript can be provided on reasonable request and on agreement with the authors, to ensure that ethical and legal requirements are upheld.

**ORCID iD**
Andrea Martani http://orcid.org/0000-0003-2113-1002

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
