## [Reviewer comments · BMJ Open]

ARTICLE DETAILS

TITLE (PROVISIONAL)	"It's not something you can take in your hands". Swiss experts' perspectives on health data ownership: an interview-based study.
AUTHORS	Martani, Andrea; Geneviève, Lester; Elger, Bernice; Wangmo, Tenzin

VERSION 1 – REVIEW

REVIEWER	Rebecca D Kush, PhD Catalysis Research USA
REVIEW RETURNED	21-Nov-2020

GENERAL COMMENTS	This is a very interesting topic and there is information from your study that may contribute to the field. However, the manuscript would benefit from a number of revisions. Suggestions are: a) The context seems to be too narrow since 'ownership' of data becomes more relevant as there are requests or initiatives to share it. There are many references on Data Sharing that should be at least cited and brought into the background and discussion sections; these include but are not limited to the CORBEL initiative that is described in BMJ Open December 2017 and the National Academies of Science Workshops, the most recent being November 2019. b) The current references are heavily weighted towards the legal angle, and it may be good to point this out. It seems that there should be a definition of 'ownership' cited somewhere in this manuscript and it is somewhat surprising that this was not part of the survey. Perhaps this can be explained in the article? c) There should probably be more context provided in relationship with the GDPR and how this deals with data ownership. d) The strengths and limitations section in the abstract and the limitations that are in the section after the Discussion are not aligned; they should be. e) It would be good to point out more about what makes Switzerland different than other countries in this regard and/or how the results in this study can benefit other countries. f) The idea of data as a commodity with value is not new or surprising; this has been around for quite some time, especially in other fields such as online marketing. g) It is not clear to this reviewer what 'data cooperatives' are; this could be explained better to help the reader comprehend. h) The discussion around whether data 'processors' become owners should elaborate on what the processing is for (secondary use). If this is for research, then the concept of 'consent' or 'informed consent' is quite relevant since the patients agree to share their data for these purposes. Consent should be discussed in this manuscript. i) There are minor grammatical errors throughout the paper, e.g. missing commas and the second bullet in the first strengths and limitations
---

	section. Also the word 'contribution' seems out of place, but perhaps that is what BMJ Open uses for a manuscript or report?
--	--

REVIEWER	Yelena Gorina CDC, USA
REVIEW RETURNED	25-Nov-2020

GENERAL COMMENTS	Dear Authors of the manuscript ““It’s not something you can take in your hands”. Swiss stakeholders’ perspectives on health data ownership...” I greatly appreciate an opportunity to learn about your research and to review your manuscript on an important subject of the health data ownership and offer my comments listed below. Overall comments  • Figure 1 describes main themes of the semi-structure questionnaire concerning data ownership. For better clarification of the options for answers and responses by the stakeholders, suggest including an appendix with relevant parts of the questionnaire. • The manuscript raises an important question of who owns the data – data processor or the patients, however stakeholders representing patients were not included in the study. Participants, such as Swiss Cystic Fibrosis Association (CFCH) or Swiss AIDS Federation, may help the readers to assess patients view on patient’s records ownership and sharing, especially records with a sensitive medical information. • From the paper and Figure 1, it seems that the options for the health data owner suggested to the study participants was either the data processors or patients. European law on data protection introduces also a data controller. By definition, “ The data controller determines the purposes for which and the means by which personal data is processed... The data processor processes personal data only on behalf of the controller” (https://ec.europa.eu/info/law/law-topic/data-protection/reform/rules-business-and-organisations/obligations/controller-processor/what-data-controller-or-data-processor_en). Suggest including an explanation why data controllers (persons or organizations) were not included as an option for data owner or mention it as a study limitation. • Suggest to include in the discussion how findings relate to the Swiss Federal Act on Data Protection (235.1, available on the portal of the Swiss government (https://www.admin.ch/opc/en/classified-compilation/19920153/index.html)). As sensitive information, health data seems to be a subject of the law. • Suggest including in the discussion an issue of the epidemic response data ownership, when the public health authorities and the government need to use the data for health emergency response. Other comments  • Section 2.4 Data collection, line 119. “The interviews were conducted independently by either LDG or AM, two male PhD candidates ... During one interview by AM, a female PhD student...” Suggest delete gender identification of the researchers as irrelevant. • Section 2.5 Data analysis, line 143. “We used MAXQDA.18 as an analytical software...” Suggest including a short description of MAXQDA.18 or a reference to the software description. • Section 3.2 Uncertainties about data ownership, line 211. “...tension that exists between the fact
--

	that the data somehow is theoretically related to patients...” Suggest deleting the word “somehow” and “theoretically” as revealing sensitive information for identifiable patient may have real consequences, e.g. stigma.  • Discussion. “It might be that the fact that patients must often be asked for consent before their data is used...” Suggest clarifying if giving consent is required by the law as in some countries, or not.
--	---

REVIEWER	Shona Kalkman University Medical Center Utrecht, the Netherlands
REVIEW RETURNED	25-Nov-2020

GENERAL COMMENTS	Thank you for the opportunity to review this original and well-written paper. The topic is very relevant; 'data ownership' is mentioned time and time again in discussions about responsible health data research, often as a necessary condition, yet it is rarely defined as a concept. As the authors (rightly) hypothesize, people may hold different interpretations of the concept, which has consequences for how to operationalize it. The research question sets out to ask stakeholders in a qualitative study about their views in order to get a better grip on the concept. The study design is robust and well-described and the results are structured and presented in a clear fashion. In the Discussion section, the authors provide a nuanced reflection on the major findings and list a number of important implications. Overall, I think this paper really adds new insights (both empirical and conceptual) to the existing body of literature on the issue of responsible data sharing. I just have a few points for the authors to consider:  - the paper talks about 'stakeholders' in a quite general sense. When I read the the title and abstract I first assumed the paper would also deal with patient and public views, given they're major stakeholders, or ethics committees. It was only a bit into the paper that I realized this was not the case. I wouldn't go so far to say it's a shortcoming not to have included patients and the public (quite opposite, I think the sample ultimately is well-selected), but it does raise the question whether there was a particular reason to focus on the specific stakeholders that were interviewed. My suggestion would be to quite early on explain that you focus on specific stakeholders and not others, and why. It doesn't need to be elaborate, and definitely keep the title as it is for brevity. - The reflection the authors provide in the Discussion section on the results is quite interesting, especially when they talk about ownership in the context of 'property' and as described in terms of market and trade. I wonder whether the analogy of data with physical property (like organs) holds properly, as respondents mention that 'it's not something you can hold in your hands'. Data can be re-used over and over again, and be shared with a massive amount of parties at the same time. You may consider this as a defense against the idea of data as property that is part of a transaction between two parties. - You might want to run through your manuscript one more time to check for typo's. I found at least two (line 95 and line 356).
---

VERSION 1 – AUTHOR RESPONSE

Reviewer: 1

Comments to the Author

This is a very interesting topic and there is information from your study that may contribute to the field. However, the manuscript would benefit from a number of revisions.

Suggestions are: a) The context seems to be too narrow since 'ownership' of data becomes more relevant as there are requests or initiatives to share it. There are many references on Data Sharing that should be at least cited and brought into the background and discussion sections; these include but are not limited to the CORBEL initiative that is described in BMJ Open December 2017 and the National Academies of Science Workshops, the most recent being November 2019.

RESPONSE

We thank the reviewer for this comment, which has prompted us to better collocate our investigation into the topic of health data sharing. We included a reference to the Corbel initiative and the recommendation on data governance that have been developed therefrom. We also mentioned the Proceedings of one workshop by the National Academies of Science that tackles these topics and provides a good overview of the *status quo* of the debate on medical data sharing in the United States – also showing how the issue of finding appropriate governance is central also in the US. Such modifications can be found at lines 42-48.

b) The current references are heavily weighted towards the legal angle, and it may be good to point this out. It seems that there should be a definition of 'ownership' cited somewhere in this manuscript and it is somewhat surprising that this was not part of the survey. Perhaps this can be explained in the article?

RESPONSE

We agree with the reviewer that it might be useful to make readers aware that a good part of the literature presented in the introduction is of a legal nature, since a considerable part of the debate on medical data ownership has involved the legal community. We made sure to highlight this at lines 59-60, where we also provided additional references beyond the legal domain, thus confirming that the debate on data ownership spans broader.

A potential cross-disciplinary general definition of ownership is provided at lines 306-308. We also added some lines earlier in the manuscript explaining that a shared definition of data ownership lacks, but that the debate on ownership centers primarily on the two issue of controlling data and benefitting from them (see lines 57-59).

In the conclusion, we also insisted on the necessity – due to the different understandings that stakeholders have of ownership – to try and reach a common definition of the concept and its implications. We then specified that – in case the language of ownership proves impracticable or too divisive for the context of health data – alternative governance paradigms should be promoted (such as stewardship), see lines 445-452.

c) There should probably be more context provided in relationship with the GDPR and how this deals with data ownership.

RESPONSE

In accordance with the useful recommendation by the reviewer, we have added some more information to the section where we had already briefly referred to the GDPR and how this deals with data ownership (see lines 64-70). Moreover, we have added other details about the GDPR throughout the text during the revision (see lines 354ss).

d) The strengths and limitations section in the abstract and the limitations that are in the section after the Discussion are not aligned; they should be.

RESPONSE

We thank the reviewer for pointing out this discrepancy, which we took care to amend. We now mentioned after the abstract a more representative description of the limitations, which we then illustrate in greater details within the appropriate section of the manuscript.

e) It would be good to point out more about what makes Switzerland different than other countries in this regard and/or how the results in this study can benefit other countries.

RESPONSE

This helpful comment by the reviewer prompted us to add some more general considerations in the section "Conclusion", which also show how the findings of our study can be relevant for other countries (see lines 435-443). Moreover, in line with a previous comment by the reviewer, we also added references to international initiatives which are trying to establish appropriate forms of health data

governance, to show that the premises of our study lie in an issue (improving data governance) which is of relevance for many countries.

f) The idea of data as a commodity with value is not new or surprising; this has been around for quite some time, especially in other fields such as online marketing.

RESPONSE

We agree that the idea of conceiving data as a commodity with value has been circulating in the literature (both scientific and not) for some time. We specified this at line 398-399.

We still repute this is a relevant finding within our study, since it confirms that the conception of data as a commodity has definitely established itself, and must – in our view – be a fundamental part of the discussion around the topic of how to regulate the control over data and the benefit that can derive from data sharing. We enriched our discussion on this topic at lines 406-417 and also at lines 447-452, by exposing the problem that – until data are conceived as a precious resource – there will be a natural tendency of data-controllers to try and exclude others from accessing them.

g) It is not clear to this reviewer what 'data cooperatives' are; this could be explained better to help the reader comprehend.

RESPONSE

We agree with the reviewer that the concept of 'data-cooperatives' can be more thoroughly explained. We thus added a longer explanation at lines 345-348 and also a further reference where readers might find additional reflections on this topic.

h) The discussion around whether data 'processors' become owners should elaborate on what the processing is for (secondary use). If this is for research, then the concept of 'consent' or 'informed consent' is quite relevant since the patients agree to share their data for these purposes. Consent should be discussed in this manuscript.

RESPONSE

We agree that a few more words on the role of informed consent with respect to medical data use (and re-use/secondary use) and on how consent allows patients to (partially) control their data are to be added. We thus included a few reflections on this topic at lines 353-367, with additional references that better contextualize the cases where data (secondary) processing can happen without consent, thus reducing the control of the individual on their data and increasing data-processors (e.g. researchers) control.

i) There are minor grammatical errors throughout the paper, e.g. missing commas and the second bullet in the first strengths and limitations section. Also the word 'contribution' seems out of place, but perhaps that is what BMJ Open uses for a manuscript or report?

RESPONSE

We carefully proofread the paper and amended the grammatical slip-ups and other typos that were present. Moreover, we avoided the word "contribution" and consistently used "paper" or "manuscript" throughout our writing. We thank Reviewer 1 for prompting us to focus on these elements, something which benefits the readability of the manuscript.

Reviewer: 2

Comments to the Author

Dear Authors of the manuscript "It's not something you can take in your hands". Swiss stakeholders' perspectives on health data ownership..."

I greatly appreciate an opportunity to learn about your research and to review your manuscript on an important subject of the health data ownership and offer my comments listed below.

Overall comments

- Figure 1 describes main themes of the semi-structure questionnaire concerning data ownership. For better clarification of the options for answers and responses by the stakeholders, suggest including an appendix with relevant parts of the questionnaire.

RESPONSE

We thank the reviewer for this comment, which has prompted us to provide additional details on the sort of questions that were asked during our study. Instead of providing an appendix with the relevant parts of our interview guide, we inserted some references to it directly in the text, so that readers can readily have an overview of the questions we asked in our research (see lines 167-171).

- The manuscript raises an important question of who owns the data – data processor or the patients, however stakeholders representing patients were not included in the study. Participants, such as Swiss Cystic Fibrosis Association (CFCH) or Swiss AIDS Federation, may help the readers to assess patients view on patient's records ownership and sharing, especially records with a sensitive medical information.

RESPONSE

We agree with the reviewer that the perspective of patients on data control and ownership are of noteworthy relevance, since also patients (and patient organizations) can be considered relevant stakeholders. We thus better explained the rationale behind our decision to NOT include patients directly in the project in which this study is nested, at lines 129-133. We also explained that we attempted to explore patients' perspectives "indirectly" by contacting a director of a patient organization who also satisfied our other criteria for enrollment. However, the contacted representative declined the invitation to participate.

- From the paper and Figure 1, it seems that the options for the health data owner suggested to the study participants was either the data processors or patients. European law on data protection introduces also a data controller. By definition, "The data controller determines the purposes for which and the means by which personal data is processed... The data processor processes personal data only on behalf of the controller" (https://ec.europa.eu/info/law/law-topic/data-protection/reform/rules-business-and-organisations/obligations/controller-processor/what-data-controller-or-data-processor_en). Suggest including an explanation why data controllers (persons or organizations) were not included as an option for data owner or mention it as a study limitation.

RESPONSE

We added a reference to the data controllers and data processors in EU law at lines 66-68, as recommended by the reviewer. However, we did not consider extensively such distinction in our results as this differentiation (data controller vs data processor) is currently not present in Switzerland, although it will be introduced in the future with the reform to the Federal Act on Data Protection - a mention of which we now added at lines 318-320 – which will come into force only in 2022. To better clarify this, we also added a footnote to the text (see new footnote number 1).

- Suggest to include in the discussion how findings relate to the Swiss Federal Act on Data Protection (235.1, available on the portal of the Swiss government (<https://www.admin.ch/opc/en/classified-compilation/19920153/index.html>)). As sensitive information, health data seems to be a subject of the law.

RESPONSE

We thank the reviewer for prompting us to reference the Swiss Federal Act on Data Protection. We added a comment on such law and on its current revision, which has however not implemented – as one initiative in the Swiss parliament was proposing – a 'legal' right to data ownership. See lines 317-320 and 354-357.

- Suggest including in the discussion an issue of the epidemic response data ownership, when the public health authorities and the government need to use the data for health emergency response.

RESPONSE

This useful comment by the reviewer inspired us to draw a relevant connection between the topics of data ownership, pandemic response and secondary use of data, with a particular reference to a political initiative that was promoted in Switzerland during the COVID crisis (see lines 414-417).

Other comments

- Section 2.4 Data collection, line 119. "The interviews were conducted independently by either LDG or AM, two male PhD candidates ... During one interview by AM, a female PhD student..." Suggest delete gender identification of the researchers as irrelevant.

RESPONSE

We eliminated the gender identification as suggested with respect to the segment "During one interview by AM...", which was describing the context of one of the interviews.

We however prefer to keep the gender identification in the other segment (i.e. "the interviews were conducted independently by either LDG or AM, two male PhD candidates), since for this study we followed COREQ-32 (as also highly recommended by BMJopen policies), which require to also refer to the gender of the researchers (Item number 4 in the COREQ-32 guidelines, mentioned in section 2.1 of the manuscript).

- Section 2.5 Data analysis, line 143. “We used MAXQDA.18 as an analytical software...” Suggest including a short description of MAXQDA.18 or a reference to the software description.

RESPONSE

We provided the reference to an extensive resource that explain in great details the functioning of MAXQDA, as recommended by the reviewer (see new reference number 42)

- Section 3.2 Uncertainties about data ownership, line 211. “...tension that exists between the fact that the data somehow is theoretically related to patients...” Suggest deleting the word “somehow” and “theoretically” as revealing sensitive information for identifiable patient may have real consequences, e.g. stigma.

RESPONSE

We agree with the reviewer and cancelled the two words, see line 234.

- Discussion. “It might be that the fact that patients must often be asked for consent before their data is used...” Suggest clarifying if giving consent is required by the law as in some countries, or not.

RESPONSE

We agree with the point that the role of (patients’) consent for the use of their data should be added, as also pointed out by Reviewer 1. We thus added a few reflections on consent and on its role at lines 353-367.

Reviewer: 3

Comments to the Author

Thank you for the opportunity to review this original and well-written paper. The topic is very relevant; ‘data ownership’ is mentioned time and time again in discussions about responsible health data research, often as a necessary condition, yet it is rarely defined as a concept. As the authors (rightly) hypothesize, people may hold different interpretations of the concept, which has consequences for how to operationalize it. The research question sets out to ask stakeholders in a qualitative study about their views in order to get a better grip on the concept. The study design is robust and well-described and the results are structured and presented in a clear fashion. In the Discussion section, the authors provide a nuanced reflection on the major findings and list a number of important implications. Overall, I think this paper really adds new insights (both empirical and conceptual) to the existing body of literature on the issue of responsible data sharing. I just have a few points for the authors to consider:

- the paper talks about ‘stakeholders’ in a quite general sense. When I read the the title and abstract I first assumed the paper would also deal with patient and public views, given they’re major stakeholders, or ethics committees. It was only a bit into the paper that I realized this was not the case. I wouldn’t go so far to say it’s a shortcoming not to have included patients and the public (quite opposite, I think the sample ultimately is well-selected), but it does raise the question whether there was a particular reason to focus on the specific stakeholders that were interviewed. My suggestion would be to quite early on explain that you focus on specific stakeholders and not others, and why. It doesn’t need to be elaborate, and definitely keep the title as it is for brevity.

RESPONSE

We agree with the reviewer that it might be appropriate to expand the explanation on how choice of stakeholders and its rationale. We did this at the end of section 2.3, at lines 129-133. Moreover, for further improving clarity we substituted the word “stakeholders” with the more precise word “experts” (which more specifically represents the kind of stakeholders we considered) in the title and throughout the manuscript.

- The reflection the authors provide in the Discussion section on the results is quite interesting, especially when they talk about ownership in the context of ‘property’ and as described in terms of market and trade. I wonder whether the analogy of data with physical property (like organs) holds properly, as respondents mention that ‘it’s not something you can hold in your hands’. Data can be re-used over and over again, and be shared with a massive amount of parties at the same time. You may consider this as a defense against the idea of data as property that is part of a transaction between two parties.

RESPONSE

We thank you for this rich comment, which has prompted us – together with another comment by Reviewer 1 – to deepen our reflection in this section of the discussion.

We explained that it is commonly supposed that “proptertising data” (i.e. assigning clear ownership rights in data) would favour their “marketization”. However, this causal relationship is not certain. First, our interviewers spoke about data in market terms even if data are not fully “proptertised” in Switzerland (i.e. there are no well-defined ownership rights in data). Second, other authors have showed that it is the lack of clear ownership rules which can actually cause the development of an imbalanced data-market. Overall we thus developed our arguments around these issues between lines 397 and 417).

We also considered the valid counter-argument that you proposed in your comment and wrote that “Indeed, although data can be re-used over and over again, and shared with a massive amount of parties at the same time, the fact remains that “the tricky part is in getting to the source of [data], i.e. people, in the first place” (see lines 410-412).

- You might want to run through your manuscript one more time to check for typo's. I found at least two (line 95 and line 356).

RESPONSE

We thank the reviewer for spotting the two typos. We made sure to correct them and to carefully check the rest of the manuscript.

VERSION 2 – REVIEW

REVIEWER	Yelena Gorina NCHS, USA
REVIEW RETURNED	03-Feb-2021

GENERAL COMMENTS	Dear Authors of the manuscript ““It’s not something you can take in your hands”. Swiss stakeholders’ perspectives on health data ownership...”, Thank you for addressing the previous comments and I greatly appreciate an opportunity to review your manuscript. I have only few minor comments to avoid possible confusion by the reader and add to the clarity of the paper. Line 49. “ Even single research institutions - like the UK Biobank - have thus dedicated particular attention to the ethics and governance of the data they manage.[4]” - delete “thus”. Line 130. “Our sample did not include patients or members of the public, since we wanted to focus on those stakeholders with first-hand experience in the management and use of medical databases.” – delete “those.” Line 136. “The interviews were conducted independently by either LDG or AM, two male PhD candidates with training on data collection and qualitative research methods. During one interview by AM, another female PhD student came along as an observer to gather experience on the conduction of interviews, upon agreement of the interviewee.” _ Is mention of the interviewee’ sex relevant? If it is, please explain why; if not, I suggest to deleting “male” and “female” mentions. Line 208. “Only some experts seemed to suggest that ownership by the patient has to do with ‘empowering’ them (E4 in table 1).” – Use capital “T” in “Table 1.”
---

	Page 34, footnote. “The distinction between data-controller and data-processor will only be introduced in the next future by the reform of the Swiss Federal Act on Data Protection, which will come into force in 2022 and which is referenced in the Discussion.” – suggest revising the statement, e.g. “...will be introduced in the near future by the reform of the Swiss Federal Act on Data Protection, which will come into force in 2022 and is referenced in the Discussion.” Page 35, footnote. “So, for example 23REngP means: 23rd interview, with a Researcher, conducted in English and via Telephone.” – According to the explanation, the interview was conducted in Person, not via Telephone. Also, suggest dropping the word “So”. Line 318. “In the Swiss parliament, there was an effort to modify the constitution to declare personal...” – Use capital “C” in the work “Constitution”. Line 401. “... since health data – like biological material and organs – are something intimate and connected to the concept of human dignity.” – I doubt that biological materials and organs can be considered as data. Suggest changing to “... since health data – such as characteristics of the biological material and organs – are ...” Line 437. “The findings of this study may thus be of use not only for Switzerland, but also for all other countries which are trying to implement ...” - Suggest to delete the word “all”.
--	--

VERSION 2 – AUTHOR RESPONSE

Reviewer 2

Comments to the Author:

Dear Authors of the manuscript ““It’s not something you can take in your hands”. Swiss stakeholders’ perspectives on health data ownership...”, Thank you for addressing the previous comments and I greatly appreciate an opportunity to review your manuscript. I have only few minor comments to avoid possible confusion by the reader and add to the clarity of the paper.

RESPONSE

We would like to thank the reviewer for the precious and constructive review both now and in the previous round of review. With your latest comment, we have had the chance to improve the clarity of the manuscript.

Line 49. “ Even single research institutions - like the UK Biobank - have thus dedicated particular attention to the ethics and governance of the data they manage.[4]” - delete “thus”.

RESPONSE

We implemented the recommended change. See line 49.

Line 130. “Our sample did not include patients or members of the public, since we wanted to focus on those stakeholders with first-hand experience in the management and use of medical databases.” – delete “those.”

RESPONSE

We agree that the “those” can be confusing and thus cancelled it. See line 130.

Line 136. “The interviews were conducted independently by either LDG or AM, two male PhD candidates with training on data collection and qualitative research methods. During one interview by AM, another female PhD student came along as an observer to gather experience on the conduction of interviews, upon agreement of the interviewee.” _ Is mention of the interviewee’ sex relevant? If it is, please explain why; if not, I suggest to deleting “male” and “female” mentions.

RESPONSE

We mentioned gender of AM and LDG (the interviewers) to comply with the COREQ 32 guidelines for reporting qualitative research, which recommend to mention the gender of the interviewers (item 4 of COREQ, see also reference 37 in the manuscript). We would thus like to maintain that indication. We agree that the reference to the gender of the PhD student who came along for one interview can be cancelled. See line 138.

Line 208. “Only some experts seemed to suggest that ownership by the patient has to do with ‘empowering’ them (E4 in table 1).” – Use capital “T” in “Table 1.”

RESPONSE

We have corrected this slip-up as rightly recommended. See line 209.

Page 34, footnote. “The distinction between data-controller and data-processor will only be introduced in the next future by the reform of the Swiss Federal Act on Data Protection, which will come into force in 2022 and which is referenced in the Discussion.” – suggest revising the statement, e.g. “...will be introduced in the near future by the reform of the Swiss Federal Act on Data Protection, which will come into force in 2022 and is referenced in the Discussion.”

RESPONSE

The revised statement suggested by the reviewer reads better, hence we implemented it in the manuscript. See footnote 1.

Page 35, footnote. “So, for example 23REngP means: 23rd interview, with a Researcher, conducted in English and via Telephone.” – According to the explanation, the interview was conducted in Person, not via Telephone. Also, suggest dropping the word “So”.

RESPONSE

We thank the reviewer for picking up this mistake, which we cared to amend. See footnote 2.

Line 318. “In the Swiss parliament, there was an effort to modify the constitution to declare personal...” – Use capital “C” in the work “Constitution”.

RESPONSE

We corrected as indicated and wrote Constitution with “C” as a capital letter. See line 317.

Line 401. “... since health data – like biological material and organs – are something intimate and connected to the concept of human dignity.” – I doubt that biological materials and organs can be considered as data. Suggest changing to “... since health data – such as characteristics of the biological material and organs – are ...”

RESPONSE

The reviewer points out a particularly interesting and controversial matter. One of the major experts in data regulation in Europe has reflected on this exact matter in a past publication (see “Bygrave LA. The body as data? Biobank regulation via the ‘Back Door’ of data protection law. *Law, Innovation and Technology* 2010;2(1):1-25). Since this tackling this debate more goes beyond the scope of our manuscript, we thus modified our sentence along the lines of the suggestion by the reviewer, in order to avoid potential misunderstanding. See line 400.

Line 437. “The findings of this study may thus be of use not only for Switzerland, but also for all other countries which are trying to implement ...” - Suggest to delete the word “all”.

RESPONSE

We agree that it is more appropriate to write as the reviewer recommended and we thus eliminated the word “all”. See line 438.